# Technological Requirements and Challenges in Wireless Body Area Networks for Health Monitoring: A Comprehensive Survey

**DOI:** 10.3390/s22093539

**Published:** 2022-05-06

**Authors:** Lisha Zhong, Shuling He, Jinzhao Lin, Jia Wu, Xi Li, Yu Pang, Zhangyong Li

**Affiliations:** 1School of Communication and Information Engineering, Chongqing University of Posts and Telecommunications, Chongqing 400065, China; zhonglisha@swmu.edu.cn (L.Z.); heshuling11@163.com (S.H.); linjz@cqupt.edu.cn (J.L.); wujia@swmu.edu.cn (J.W.); lixixyz@126.com (X.L.); pangy@cqupt.edu.cn (Y.P.); 2School of Medical Information and Engineering, Southwest Medical University, Luzhou 646000, China; 3School of Bioinformatics, Chongqing University of Posts and Telecommunications, Chongqing 400065, China

**Keywords:** wireless body area networks, application-specific integrated circuit, security and privacy, energy efficiency, health monitoring

## Abstract

With the rapid growth in healthcare demand, an emergent, novel technology called wireless body area networks (WBANs) have become promising and have been widely used in the field of human health monitoring. A WBAN can collect human physical parameters through the medical sensors in or around the patient’s body to realize real-time continuous remote monitoring. Compared to other wireless transmission technologies, a WBAN has more stringent technical requirements and challenges in terms of power efficiency, security and privacy, quality of service and other specifications. In this paper, we review the recent WBAN medical applications, existing requirements and challenges and their solutions. We conducted a comprehensive investigation of WBANs, from the sensor technology for the collection to the wireless transmission technology for the transmission process, such as frequency bands, channel models, medium access control (MAC) and networking protocols. Then we reviewed its unique safety and energy consumption issues. In particular, an application-specific integrated circuit (ASIC)-based WBAN scheme is presented to improve its security and privacy and achieve ultra-low energy consumption.

## 1. Introduction

The prevalence of chronic diseases such as hypertension, diabetes and obesity is currently aggravating the burden on public-funded healthcare systems and causing many economic and social challenges in some countries. According to the World Health Organization (WHO), the global population over the age of 60 will reach about 2.1 billion by 2050 [1]. These public healthcare problems are exacerbated by the rapid growth of the elderly population, as elderly people are more prone to suffer from chronic diseases. Chronic diseases constitute a major portion of human health risks, accounting for more than two-thirds of all deaths worldwide [2]. Cardiovascular disease accounts for 30 percent of all deaths. Globally, more than 180 million people have diabetes, which is estimated to affect about 360 million people by 2030 [3,4]. In 2015, more than 2.3 billion people were overweight, the leading cause of these chronic diseases. Moreover, the rapid rise in debilitating neurodegenerative diseases, such as Alzheimer’s and Parkinson’s, threatens millions of people. These diseases are not simply the result of an aging population, but are caused by sedentary behavior, inappropriate eating habits and insufficient physical activity [5,6]. Many studies [7,8,9] have demonstrated that chronic diseases can be effectively prevented at an early stage, suggesting that early diagnosis and detection are vitally important. Therefore, it is of great and urgent significance to realize real-time monitoring of human health via disease surveillance and health evaluation. 

Due to the current developments in sensors and wireless communication technology, wireless body area networks (WBANs) may alleviate or even solve the problems of rampant chronic diseases, an aging population, a shortage of medical facilities, etc. A WBAN is a body network that enables communication between people and things by connecting nodes with sensors in, on or around humans [10]. Data transmission between these nodes is limited to an ultra-short distance of 2 m by wireless means. Figure 1 illustrates the basic idea of the WBAN and its applications. The responsibility of each node is to collect physiological parameters, such as electrocardiogram (ECG), electroencephalogram (EEG), blood oxygen saturation (SpO_2_), blood pressure (BP) and heart rate variability (HRV). The terminal plays a role of a personal server to gather all the data from nodes and then transmit them to the Internet. Moreover, WBAN system can provide bio-feedback to the patients from the remote servers. The servers on the remote system cannot only process the data efficiently, but also provide some services, such as real-time monitoring and health consultation, which is helpful for the management of chronic diseases. As shown in Figure 1, WBAN has a huge market, from which equipment manufacturers, operators, solution providers and service providers can all take a profit. That is another important reason why WBANs attracted great attention around the world as soon as the idea emerged. Taking advantage of WBAN technology is important for economic development as a new growth engine. Although WBANs are attractive for many applications, they are still in their infancy, and this new wireless technology combines multiple disciplines, such as communication, bio-engineering and microelectronics, making it difficult to solve the key issues.

Figure 2 describes the communication architecture of a WBAN health monitoring system. WBANs can be categorized into three tiers of level communication: intra-WBAN, inter-WBAN, and beyond-WBAN communications [11]. The tier-1 is intra-WBAN communication, which consists of a set of sensors placed on or implanted into the human body. Sensors preserve star topology and are connected to a centralized node in this tier with the function of collecting and transmitting various human physiological parameters. This communication is only between the sensors and the sink, whose task here is to process the collected information and transmit it to Tier-2 through ZigBee, Bluetooth, Wi-Fi or some other short-distance transmission technology. Tier-2 is inter-WBAN communication, which uses smartphones, personal computers or other intelligent electronic devices. Ad hoc architecture is distributed to communicate in this tier with a random topology. The function of Tier-2 is to forward the information sent by the sensor to Tier-3 (terminal center) through 3G/4G/5G, WLAN and other wireless technologies. Tier-3 is the beyond-WBAN communication, in which the terminal center is mainly composed of remote servers. The function is to store and analyze the received information, which can be used for monitoring, diagnosis and treatment of diseases. Especially when the received data are abnormal, an emergency response and alarm can be initiated, contributing to speeding up emergency care. Each tier has special technical difficulties and requirements that need to be solved. Numerous papers [12,13,14,15,16] have summarized the challenges faced by WBANs, but these surveys, which focus on some specific aspects of WBANs, are limited to systematically describing WBANs and only refer to several technical points, failing to provide solution schemes using integrated circuits (IC). In this paper, we denote the application and technical requirements of WBAN, and give the problems systemic, including the sensor nodes, frequency bands, channel models, ultra-short range communication systems, networking and protocols, safety and privacy protection and energy consumption. For each problem, we present a comprehensive and deep analysis and try to present some hints for technical solutions. In order to solve the problems of safety and energy consumption in particular, we put forward designs for application-specific integrated circuit (ASIC) schemes which not only achieved good performances but also can contribute to the future batch production and application of WBANs. 

## 2. WBAN Applications for Health Monitoring

WBAN technology is widely used in health monitoring, military, sports, entertainment, aerospace and many other fields involving human beings, presenting huge economic benefits and social value. As shown in Table 1, it is commonly divided into medical and non-medical applications [13]. WBANs could contribute to protecting those people who may be exposed to life-threatening conditions, such as firefighters, soldiers, deep-sea explores and space explorers. Examples include providing firefighters with monitoring of special environments, such as fire sites and areas with toxic gases; providing emergency early warnings to improve the safety of firefighters; monitoring the physical condition of athletes in real time to set appropriate training intensity; and wirelessly transmitting vital military information to control centers or remotely commanding units. The non-medical applications can be categorized as real-time streaming, entertainment and non-medical emergencies. In recent years, entertainment uses, such as gesture and motion detection for games and virtual experiences, have also emerged.

Due to the huge demands of telemonitoring and telemedicine, WBANs can greatly alleviate the lack of medical resources and improve their utilization rate, making medical applications the most important applications. Medical WBAN applications could be generally subcategorized into two groups: wearable WBANs and implantable WBANs according to the sensors being on or in the human body [29] (details shown in Table 1). The advantages of ceaseless body data capture from various sensors will lead to healthcare beyond hospital limitations and facilitate exceedingly personalized and individual care at any time and any place [30]. Health monitoring to provide chronic disease surveillance and medical care services can greatly enhance the patients’ quality of life and address the medical facility shortage [31]. WBAN is capable of wireless connectivity to ECG/EEG/electromyograms (EMG); SpO_2_, BP and body temperature monitors; implanted defibrillators; sleep staging monitors; devices for high-risk pregnancy monitoring; implanted drug delivery trackers; swallowed camera pills; gait analysis systems; and emotion detection systems [32,33,34,35,36,37,38]. 

WBAN systems for health monitoring are heterogeneous and have been developed for various diseases and disabilities. Table 2 summarizes some diseases monitored using WBANs in recent studies. Currently, various types of health monitoring systems have been used for various diseases. These disease monitoring systems vary widely in terms of physiological parameters, sensor design and transmission technology. Moreover, even monitoring systems for the same disease are not uniformly standardized. Different types of disease present specific challenges for WBAN technology. Thus, health monitoring systems using WBANs are heterogeneous and have been used for various diseases and disabilities. The various disease data needed necessitate different sensors, making sensor technology an important part of WBANs. In addition, it has been found that transmission protocols commonly involve Wi-Fi, 3G/4G/5G, GPRS, Bluetooth or ZigBee, and most of them are based on smartphones [39,40,41,42]. These two parts, i.e., sensor and transmission, will be discussed in detail in later sections.

With the widespread application of WBANs in health monitoring, technical requirements need to be made specific to the types of disease and different application scenarios. We must not only have different requirements for sensor nodes and topology, but also transmission rate and real-time performance. Table 3 describes some typical technical requirements of BAN applications. Three aspects, “power consumption”, “quality of service (QoS)” and “safety for the human body” have significant differences in the BAN standards compared to 802 other standards [52]. Table 4 shows the desired ranges matching the proposed requirements of WBAN standards.

To sum up, in this chapter, we introduced some typical application scenarios of WBAN, especially summarizing various diseases monitored by using WBANs. Obviously, each disease requires selecting appropriate sensors, and WBAN technology has its own technical requirements for various diseases and application scenarios. Therefore, the WBAN is a heterogeneous system with its special technical requirements, proprietary standards and challenges compared to the traditional wireless sensor networks.

## 3. WBAN Sensor Techniques 

Sensors, which convert physical parameters into electronic signals, are the crucial components in the WBAN. Sensors have been mainly categorized into three types according to their functions [53]. Physiological sensors measure ambulatory BP, glucose (continuously), body temperature, blood oxygen, ECG, EEG, EMG, etc.; biokinetic sensors measure acceleration and the angular rate of human motion rotation; and ambient sensors measure humidity, light, sound pressure level and environmental temperature. Generally, WBAN sensors for obtaining physiological parameters of the human body are categorized into two groups: invasive (i.e., implanted sensors) and non-invasive (i.e., external sensors) [14], as shown in Table 5. Implanted sensors are inserted into the human body or under the skin with the help of surgery, whereas external sensors are directly attached to the skin or around the body.

Table 6 describes the mechanisms of several common sensors and their data rates in WBANs [54]. One important requirement is that the sensors can continuously monitor the patient’s health conditions without disturbing their activities. Since the physiological parameter signals are very weak, the correct detection and precise processing of these signals by sensor nodes is very crucial: these collected data are the basis of clinical diagnostics. In addition, sensors generate various types of data that require different processing to ensure the specific requirements are met. Types include general data, delay sensitive data, emergency data and reliability data. Due to the various data rates of sensors of human physiological parameters, how to merge multi-sensor data to describe one’s health condition in a uniform standard is difficult. To address these problems, several researchers [55,56,57] have designed novel flexible sensors to provide comfortable, soft and stretchable wearable systems for human health monitoring, enabling effective monitoring of physiological signs without affecting patients’ daily activities. Some studies have designed signal extraction schemes for each different body parameter which adapt to body status and environmental movement [58,59,60], and explored high-speed preprocessing and transmission algorithms for the weak physiological parameter signals of the human body [61,62]. There are also some studies [63,64,65] that proposed data fusion strategies for multiple sensor nodes based on different sensing mechanisms and physiological information relationships, forming a series of body parameter detection methods required for sensor node microarchitecture optimization [66,67,68].

## 4. Wireless Transmission in WBANs

Compared with other wireless transmission technology, WBANs which are placed on the human body focus on realizing short-range, low-cost, low-power and low-implementation transmission; the short-range communication channels especially are quite different. Electromagnetic waves among nodes or nodes and smart terminals would pass through the human body or spread along the body’s surface. The wireless nodes of WBAN require much lower power consumption than traditional sensor network nodes, generally with less than 1 mW peak power. WBAN communication distance is generally within 2 m, which is shorter than that of a general personal area network. These differences indicate that WBANs need to seek new wireless transmission schemes. Table 7 shows the main specifications for WBAN systems [69].

### 4.1. Frequency Bands for WBAN Channel Models

The choice of frequency bands for WBANs is an elementary factor affecting wireless communication. Table 8 provides the operation frequency bands for medical WBANs [12]. These frequency bands can be further divided into two groups (in-body or on-body) depending on the location of the sensor. The medical device radio communications frequency band (401–406 MHz) is mainly used in implantable medical devices. It has quiet channel properties and worldwide availability; it cannot be substituted by wearable sensor node frequencies. Industrial, scientific and medical (ISM) bands in the 2360–2500 MHz zone are exposed to less interference compared to the 2400–2483 MHz band [11], which is an unlicensed band occupied by Wi-Fi, Bluetooth and ZigBee as of IEEE 802.15.6. Some frequency bands are only for specific countries, such as the general telemetry (868–870 MHz) and some ISM bands (902–928 MHz) with limited bandwidth. The IEEE 802.15.6 standard presented a new communication medium, human body communications (HBC), with a frequency range of 5–50 MHz. A capacitively coupled HBC channel was developed based on the induction of the transmitter to the near electric field around the human body, and the weak coupling changes in the electric field near the human body channel are detected by the receiver. According to the application scenarios of WBANs, selecting different channel frequency bands can effectively avoid interference and solve the coexistence problem.

### 4.2. WBAN Channel Models

Various sensors are placed on the body or implanted into the body in WBAN health monitoring. Due to the complex environment of the human body and its surroundings, there are many interference signals in WBAN channels which will affect the quality of communication, directly affecting the performance of the WBAN system. WBAN channel characteristics are not only the key to constructing the network’s architecture, but also indispensable to the design of the upper-layer network protocol. Due to the complexity of human tissues and body shapes, and the diversity of environments we occupy, the difficulty of WBAN channel modeling is great. Therefore, the wireless transmission channel in the WBAN is divided into several modes, as shown in Figure 3 and Table 9. There are three typical nodes in WBANs: implanted, body surface and external nodes [76]. The channel characteristics have four main relevant factors: the frequency factor (WBAN frequency bands may include 400 MHz, 600 MHz, 2.4 GHz and UWB), environmental factors (WBAN’s environment, such as the anechoic chamber, outdoors or a hospital), antenna placement and status of human motion [77,78,79,80,81]. Environment factors mainly affect the multipath propagation caused by the surrounding complex environment. In an outdoor environment, the surface channel sees little attenuation. However, in an indoor environment, the attenuation of a WBAN channel with 900 MHz and 2.4 GHz frequency bands can be dramatic. Therefore, a perfect channel model should introduce the three modifying factors to satisfy the simulation of different applications.

Differently from common wireless communication channels, which are severely affected by frequency selective fading, WBAN channels are mainly affected by flat fading. The reason is that the multipath delay is tiny due to the short distances, so the effect of multipath fading can be ignored. Concurrently, due to the variations in the surrounding environment of the human body or movement of body parts, path loss will change violently, leading to shadowing from the mean value for a given distance. The main model is a shadow fading channel and can be adjusted dynamically by three parameters—antenna position, sensor position and personal environment. Therefore, the total path loss (*PL*) is defined as the following formula [82]:(1)$$PL=PL_{0}+10n\;\log_{10}\left(\frac{d}{d_{0}}\right)+S$$
where *PL*_0_ is the path loss at a reference distance *d*_0_, *d* is the distance between the antennas, *n* represents the path-loss exponent and *S* represents the shadow.

The first parameter is affected by the antennas’ size, shape and transmitting direction; the second parameter relies on its position in the human body; and the third parameter is based on statistical modifications for height, weight and gender. Furthermore, various environmental models should be established, such as family chambers, hospitals and outdoors, which have different impacts on path loss and fade. Therefore, the environment in which WBAN communication occurs is highly dynamic and unstable, and a specific communication channel must be considered to guarantee good communication performance in a highly dynamic and unstable environment [83]. The IEEE standard channel models are only static, having no time varying effects and correlation features. Several dynamic WBAN channel models have been proposed, as shown in Table 10.

### 4.3. Physical (PHY) Layer and Medium Access Control (MAC) Layer

The IEEE 802.15.6 standard defines new PHY and MAC layer specifications for WBANs, providing ultra-low-power, low-cost and short-range wireless communication which operates in or around the human body. IEEE 802.15.6 standard outlined three different PHY schemes: narrowband (NB), UWB and HBC. NB and UWB PHY are based on radio frequency (RF) propagation, and HBC is based on a new non-RF technique. NB PHY included seven frequency bands from 402 to 2483.5 MHz for a total of 230 channels, of which the 402–405 MHz band is used for implantable devices, three different frequency bands (863–956 MHz) are used for wearable applications and 2360–2400 MHz is used for medical demands. UWB PHY included 11 frequency bands ranging from 3494.4 to 9984.0 MHz. Due to the high signal attenuation and severe shadow effect through the human body, these radio frequency bands are not suitable for HBC. As a signal transmission medium, the frequency band of HBC ranges from 5 to 50 MHz; the center frequency band is at 21 MHz. The WBAN standard demands an ultra-short range communication mechanism limited to 2 m. As the lowest layer, PHY, determines the high-layer protocols, which requires minimizing power consumption and bit error rate (BER) [91]. Ideally, power consumption and BER increase linearly as the data rate increases from 1 kbps to 10 Mbps, resulting in consistent energy use per bit of information. In order to acquire low power consumption, the PHY signal processing algorithms at the receiver need to be designed carefully. Future works should focus on seamless connectivity in dynamic environments to minimize performance degradation in terms of data loss, latency and throughput.

WBAN is a severely resource-constrained network in which most of the power consumption is caused by the transceiver, whose duty cycle is controlled by the MAC layer. Therefore, it is very important to design an energy-saving MAC protocol to ensure efficient and reliable transmission of data using limited wireless channel resources in WBAN. The unique challenges of MAC protocol design are mainly focused on energy consumption, QoS and transmission efficiency. Early studies focused on addressing each individual problem, such as the dynamic traffic pattern [92,93] or the energy efficiency [94,95]. Liu et al. [96] proposed a TDMA-based MAC protocol that can adjust the duty cycle according to the types of nodes in the WBAN to guarantee both QoS and energy efficiency. Maman et al. [69] proposed an adaptive TDMA MAC protocol which could automatically detect the shadowing effect and adjust the superframe parameters; it provided good results in latency outage and energy consumption. Bai et al. [97] proposed a frame structure model for a self-adaptive guard band protocol based on TDMA which synchronizes the sleeping state of the nodes and the coordinator to effectively reduce the energy consumption. Lin et al. [98] proposed a MAC protocol based on channel-aware polling to optimize energy efficiency by adjusting the number of polling cycles in super frames to adapt to dynamic traffic requirements and channel fluctuations.

### 4.4. Network Protocols

WBAN is a dedicated network with special data transmission characteristics. Therefore, based on a very short-range wireless communication system, it is necessary to build new network architecture and corresponding protocols. On one hand, for the vital information, well-designed architecture optimizes network transmission [22]; on the other hand, for the body and the surface of the wireless environment, appropriate architecture optimizes protocol design [10,99]. Note that for WBAN architecture, some solutions can be obtained from wireless sensor networks. However, these results cannot be applied directly to get the best network performance because WBAN has many different characteristics. Consequently, there should be further exploration and research toward building WBAN-dedicated network protocol architecture. The main issues are listed as follows:Multi-level QoS and cross-layer optimization. In a WBAN for various types of medical applications, the network should provide different levels and types of service quality. Thus, it would be necessary to design new or improved link layer, network layer and application layer protocol, to fully guarantee the data transmission QoS with changing demands depending on the characteristics of the information needed [22,100]. In addition, the layers’ protocol for low-power design strategy and cross-layer design and optimization methods are also worthy of attention [101,102].Adaptive networking and topology control. WBANs usually consist of different types of nodes, but the node numbers of the same type are not large. The network is more focused on the different types of heterogeneous nodes in networking and service, which is one of the differences between WBANs and common WSNs [103]. Therefore, not only homogeneous nodes but also heterogeneous nodes can be supported in a self-organizing network scheme. To take the posture effects into account in the networking and managing network at the same time, the scheme requires a dynamic topology control method which is able to adapt changes to follow the physical state [10].In-network cooperation and feedback optimization. Different heterogeneous nodes cooperate with each other and complete human monitoring and information processing and transmission. This is an important feature of WBANs. To establish a dedicated WBAN architecture, it is essential to develop a collaborative framework and mechanisms between network nodes. These mechanisms include sensor-related technology of event-driven information transfer methods, sleep–wake-up mechanisms and monitoring information data fusion mechanisms [104,105]. It is worth noting that there are usually a lot of feedback loops in a WBAN which could conveniently control the reverse information transmission; thus, how to design closed-loop controlling methods and the corresponding protocol is an important issue for WBANs.Heterogeneous interconnection framework. Any one WBAN and other WBANs, personal area networks, LAN, mobile communication networks and the Internet connect together, which is affecting WBAN technology and the development of important technical factors. A heterogeneous network includes two aspects: the interconnection of heterogeneous nodes and a heterogeneous network. On one hand, a common data representation and flexible network connectivity structure should be proposed for internal heterogeneous nodes in a WBAN with the purpose of having interconnections between all kinds of sensor nodes and interconnections between nodes and gateways [106]. On the other hand, to aim at connections between the WBANs and other types of heterogeneous networks, it is necessary to build a common data communication and protocol conversion interface to complete the interconnection of WBANs and the Internet, mobile communications networks and other mainstream networks [107,108]. From the above, the former is conducive to interoperability and interconnection among WBAN devices, and the latter can provide network-level technical support for the implementation of telemedicine, which is significant for remote continuous monitoring.With the development of the Internet of Things, the increasing number of WBANs and the mobility of WBANs, interference is becoming more challenging. For a single WBAN, intra-BAN interference can be effectively avoided by using TDMA techniques, but multiple WBANs interfere with each other when they are co-located (i.e., inter-WBAN interference). Figure 4 describes the different types of interference in WBANs [109] and the parameters that cause inter-BAN interference.

### 4.5. Chapter Summary

In response to the special technical requirements of WBAN, this chapter not only introduced the IEEE 802.15.6 standard channel models, but also discussed the existing WBAN channel models in different scenarios, including the methods of channel model building approaches and their respective advantages. Finally, the three important parts of a WBAN, i.e., PHY, MAC and networking layers, were expounded on, and their respective requirements and corresponding solutions were put forward. Therefore, WBANs focus on energy consumption, bit error rate, QoS transmission efficiency, security, privacy, etc.

## 5. Security and Privacy

The human physiological parameters collected or transmitted by WBANs are highly sensitive and confidential because these determine the outcomes of clinical diagnoses and belong to the patients. With the development of Internet of Things technology, WBANs contain an increasing number of nodes, resulting in more and more critical data transmission in the networks. Therefore, security and privacy are the utmost considerations for BANs and should be guaranteed carefully to prevent information leakage and tampering. However, security schemes proposed for other networks are not suitable for WBANs due to strict resource constraints, such as energy consumption, communication rate and computing power. Hence, addressing security in WBAN environment is a vital research topic and brings additional challenges to the design of WBANs.

The IEEE 802.15.6 standard defined three security levels for WBANs, as shown in Table 11. The specific level of security is determined by the types of data and their levels of the privacy. More precisely, for level 0, the lowest level of security, no algorithm or mechanism is used during communication. At level 1, secured authentication is provided during the data transmission. Level 2 requires both authentication and encryption. This standard also included four elliptic-curve-based protocols to guarantee security. However, researchers found that those four protocols are not secure enough in some practical applications and are vulnerable to multiple attacks [110]. Therefore, based on this standard, several security schemes for WBANs have been created to enhance security in recent studies. Table 12 shows a comparison of the recent security techniques in WBANs [111].

WBAN security schemes can be mainly divided into two groups: authentication and encryption. In terms of identity authentication, the WBAN system first needs to authenticate nodes trying to join the network and check whether they have the right to access the network to prevent illegal users from intruding. Nowadays, as the application scenarios of WBANs are becoming more and more diverse, authentication schemes need to be improved according to different application requirements. In response to the security requirements of WBANs in cloud-assisted environments, Mahender Kumar and Satish Chand proposed an identity-based anonymous authentication and key agreement protocol scheme, which was proven to be secure and achieved the required security properties [119]. A three-tier security approach was presented that uses lightweight cryptography to address security in a three-layered WBAN system [122]. In order to improve the adaptability to human motion and the integrity of data transmission, a multi-hop WBAN authentication method based on a lightweight physical unclonable function was proposed [123]. Liu et al. [124] performed a certificateless signature scheme to construct two efficient remote anonymous authentication (AA) schemes for WBANs. Debiao He et al. [125] proposed a new AA scheme to avoid imitative attacks on this basis of Liu et al.’s scheme. Gangadari et al. [126] designed an improved AES algorithm for identity authentication in BAN by using one-dimensional cellular automata to replace the traditional look up table (LUT)-based S-Box method. The results showed a high level of security.

Currently, there are roughly three schemes available for BAN data encryption. The first is the key pre-distribution scheme, distributing secret keys in sensor nodes before communication. Tripathy et al. [127] described a matrix-decomposition-technique-based key predistribution approach, which provides superior key connectivity and requires less storage memory. Saikia and Hussain proposed a combinatorial-group-based key distribution method to effectively enhance the security between nodes [128]. The second is to utilize physical sign information as a means of security. Unlike other sensor networks, WBAN sensor nodes collect physiological parameters. These biometric parameters of different individuals are discrepancies. Even for the same individual, the values are not constant but change slowly over time. Thus, these physiological signals, such as EEG, naturally act as individual keys in WBANs. Seepers et al. [129] presented a heart-beat-based security scheme that extracted an inter-pulse interval feature from R waves of ECG signals and used the key generated by heart-beat interval changes to encrypt. Bai et al. [130] proposed a novel encryption method for WBANs that uses the QRS complex of the ECG signals. It has the advantages of low energy consumption, dynamic key and simple hardware implementation. Another common approach is to utilize the properties of the BAN channel as the key, which realizes a one-time key to ensure the absolute safety of the eavesdropping outside λ/2. The basic idea of BAN channel encryption is to generate the key according to the intensity of the received signal. Shen et al. [131] proposed a multi-layer authentication protocol for WBANs which generates a secure session key for relatively little in the way of computational cost. Some other complex BAN channel encryption methods include ellipse curve cryptography [132] and advanced encryption standard (AES) encryption [126]. However, there are some disadvantages, such as high consumption cost and no dynamic updating of the key in BAN channel encryption [133,134].

The above-mentioned methods focused on software implementations. The hardware implementations of these security schemes have the advantages of lower power consumption and low latency. The authors of [135] proposed a hardware implementation named ASIC to solve the security problems. It was synthesized by using SIMC 65 nm complementary metal oxide semiconductor (CMOS) technology. This ASIC contains two modules, authentication and encryption modules, and these two modules could operate together or independently, depending on the security level. Therefore, lower power consumption was achieved in some environments because of only one module operating. This proposed ASIC scheme has obvious superiority in terms of power consumption, latency and steadiness of performance.

WBAN involves human body information; therefore, security and privacy are very important, which must be considered when designing a WBAN. This section went from illustrating the security privacy levels to an overview of the existing methods for security and privacy. In addition to outlining the three schemes available for BAN data encryption in software implementations, a hardware-based ASIC scheme was also proposed.

## 6. Energy Efficiency

Compared to conventional short-range wireless communication protocols (i.e., Zigbee and Bluetooth), WBAN is varies more in energy consumption. As WBANs involve the human body, in order to make them wearable, it is often necessary to limit the size of the hardware (small sizes of 1 to 3 cm or even less), which makes it difficult to expand the battery power. Additionally, frequent battery changes are extremely inconvenient for patients, especially for the implanted devices, which can only be replaced by surgery. The hardware size and battery power limitations of a WBAN directly affect its usability and the user’s satisfaction. Therefore, energy efficiency is very important for a well-designed WBAN, and the optimization of energy consumption has become one of the current focuses of WBANs in many studies. Some studies that worked on energy harvesting included solar, vibration and thermal energy to improve energy efficiency [136,137,138,139]. Others worked on extending the battery capacity by using PHY or MAC protocols with low power consumption; some studies on the ultra-low-power IC have provided a new window for these problems and achieved good results [140,141]. Liu et al. [142] designed an ultra-low power baseband transceiver IC with 0.18 μm CMOS technology. Chen et al. [143] proposed a low-power, area-efficient baseband processor for WBAN transmitters based on the IEEE 802.15.6, which supported the specifications of the narrowband (NB) PHY by optimizing the DPSK modulator and clock gating technology. Liang et al. [144] created a hardware scheme applicable to IEEE 802.15.6, which was implemented based on FPGA and supported NB PHY, including baseband processing, radio frequency and digital-to-analog conversion. Finally, the throughput, working frequency and energy consumption of this scheme were verified. Chougrani et al. [145] proposed a baseband architecture supporting the UWB digital baseband PHY in the IEEE 802.15.6 standard, and the FPGA experiment proved that this scheme has a low packet error rate and bit error rate. Mathew et al. [146] put forward a complete NB PHY transceiver implemented in FPGA, which consists of baseband transceiver modules. An ASIC of baseband processing module in NB has been performed to solve the energy consumption of WBANs [147]. This ASIC conforms to the IEEE 802.15.6 standard. Compared to other publicized IC design schemes, this proposed ASIC showed advantages in power consumption, throughput, small area and energy efficiency. The detailed results are demonstrated in Table 13.

Designing a suitable IC for WBANs requires not only optimizing the power consumption. This is also one of the key conditions to enable WBAN to be applied on a large scale. However, many of the technologies in IC for WBAN are still in the early stage and there is no common chip for WBAN yet. In the future, the proposed ASIC can be improved by considering the cooperation between the PHY and MAC layers and the security issues.

## 7. Conclusions

As a new, rapidly spreading technology, WBANs play a significant role in health monitoring. In this paper, we have reviewed the WBAN applications in healthcare, especially those for various diseases monitoring, along with their technical requirements and challenges. Several technique issues, such as sensor nodes, wireless transmission, safety and energy efficiency, need to be seriously considered, which we discussed deeply. More importantly, in terms of the safety and energy efficiency problems, a specific ASIC was proposed to improve security and reduce energy consumption, which may be conducive to the subsequent mass promotion and application of WBANs.

## Figures and Tables

**Figure 1 sensors-22-03539-f001:**
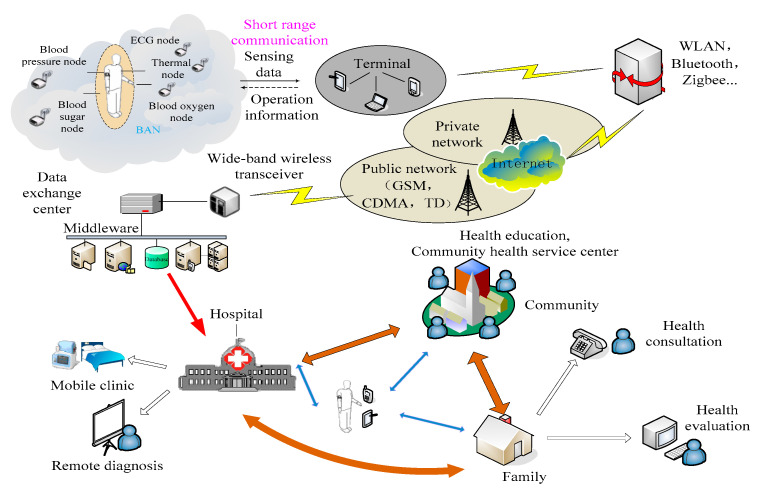
The basic idea of the WBAN system and its applications.

**Figure 2 sensors-22-03539-f002:**
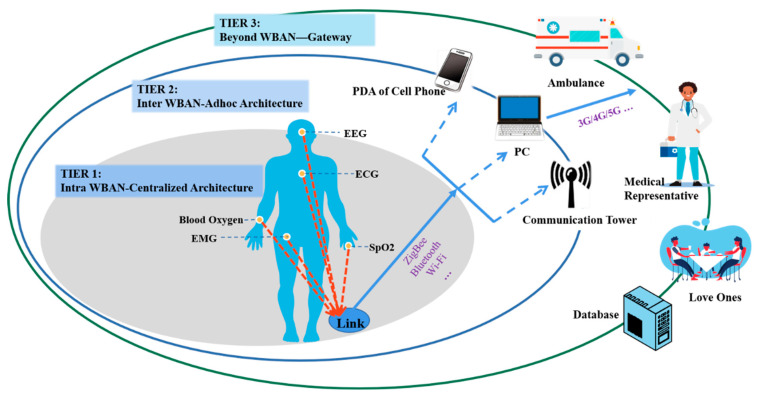
The architecture of the WBAN system.

**Figure 3 sensors-22-03539-f003:**
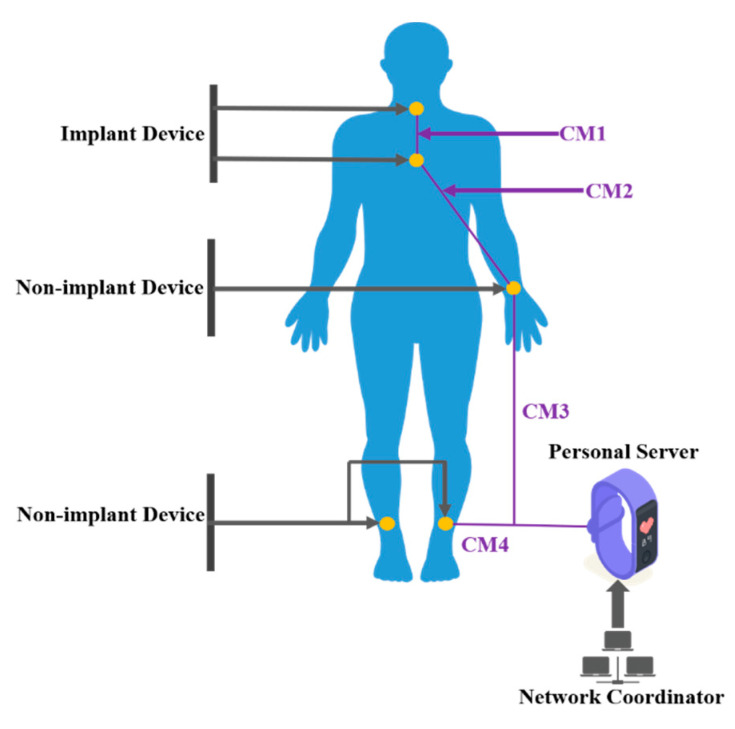
Possible communication links for body area networking.

**Figure 4 sensors-22-03539-f004:**
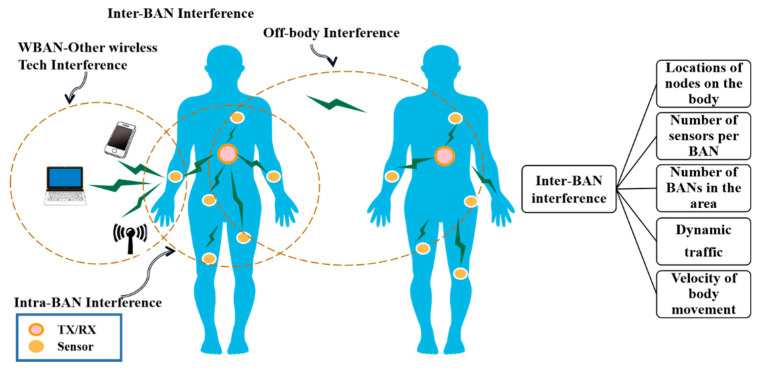
BAN interference.

**Table 1 sensors-22-03539-t001:** Some typical applications of WBANs.

Medical	Wearable WBAN	Aiding Professional and Amature Athletic Training [17] Wearable Health Monitoring [18] Asthma Monitoring [18] Sleep Staging or Monitoring [19] Fall Detection [20] Assessing Soldier Fatigue and Battle Readiness [21] Patient Monitoring [22] Telemedicine Systems [23]
	Implantable WBAN [24]	Cardiovascular Diseases [25] Diabetes Control [26] Cancer Detection [27]
Non-Medical	Real-Time Streaming [28]	Video steaming Data file transfer 3D video Sports
	Emergency (non-medical) [13]	Life-threatening conditions monitoring: firefighters, soldiers, deep-sea explorers, and space explorers.
	Entertainment Applications [17]	Games Social networking

**Table 2 sensors-22-03539-t002:** WBAN applications for monitoring various diseases.

Diseases	Collected Data	Sensor	Transmission Protocol
Depression [43]	the location, the posture, pressure→accelerometers	Barometric pressure sensor data	—
Pain assessment [44]	facial surface EMG	wearable sensor with a biosensing facial mask	hotspot of a cellphone/a smart gateway/a general router
Heart diseases [45]	BP, ECG, SpO_2_, heart rate, pulse rate, blood fat blood glucose, patients’ risk and location	ECG; blood sensing device	Bluetooth
Knees rehabilitation [39]	EMG; ECG	Accelerometer; EMG; ECG	Smartphones act as a gateway
Knee arthroplasty [40]	the angles of knee flexion	a master and slave sensor unit, the flexion angle sensor	mobile telephone network
Chronic diseases [46]	heart rate, body temperature, and blood pressure	corresponding three sensors	Bluetooth
Hypertension [47]	ECG, HRV	ECG	Bluetooth
Ubiquitous monitoring system [41]	four types of vital signs, oxygen saturation, blood pressure, heart rate, and sugar level	body sensor network	3G/Wi-Fi/Bluetooth
Cardiovascular diseases [48]	Physiological signals include ECG, BP, stress level, SpO_2_	Accelerometers; ECG	Mobile device
Heart diseases [42]	BP, pulse, body temperature, patient position, ECG	ECG; airflow; body position; BP sensor; Ambient sensors	Wi-Fi/3G/GPRS, ZigBee/Bluetooth
Diabetes [49]	blood glucose; Blood pressure; ECG	corresponding sensors	Bluetooth
Diabetes [50]	EMG, Body temperature, Heart rate, Blood pressure, Blood glucose	corresponding sensors	ZigBee
Fall detection [20]	real-time activity and fall data	motion sensors	Bluetooth
Obesity [51]	heart rate, waist circumference, physical activity, weight, glucose	chest strap; band; pedometer; pressure sensor; patch	GPRS/3G/4G/Wi-Fi

**Table 3 sensors-22-03539-t003:** Technical requirements of BAN applications.

Application	Data Rate	Nodes Number	Topology	Setup Time	P2P Latency	BER	Duty Cycle	Battery Lifetime
ECG	72 kb/s	<6	Star	<3 s	<250 ms	<10–10	<10%	>1 week
EMG	1.54 Mb/s	<6	Star	<3 s	<250 ms	<10–10	<10%	>1 week
EEG	86.4 kb/s	<6	Star	<3 s	<250 ms	<10–10	<10%	>1 week
Drug dosage	<1 kb/s	2	P2P	<3 s	<250 ms	<10–10	<1%	>24 h
Hearing aid	200 kb/s	3	Star	<3 s	<250 ms	<10–10	<10%	>40 h
Capsule endoscope	1 Mb/s	2	P2P	<3 s	<250 ms	<10–10	<50%	>24 h
Deep brain stimulation	1 Mb/s	2	P2P	<3 s	<250 ms	<10–3	<50%	>3 years
Imaging	<10 Mb/s	2	P2P	<3 s	<100 ms	<10–5	<50%	>12 h
Audio	1 Mb/s	3	Star	<3 s	<100 ms	<10–5	<50%	>24 h
temp/respiration/glucose monitor/accelerometer	<10 kb/s	<12	Star	<3 s	<250 ms	<10–10	<10%	>1 week

**Table 4 sensors-22-03539-t004:** Summary of technical requirements and their desired ranges.

Characteristic	Requirement	Desired Range
Operating distance	In, on or around the body	Typically limited in 3 m
Peak power consumption	Ultra-low	µW level in sleep mode, up to 30 mW fully active mode
Data rate	Scalable	From 1 kb/s to 10 Mb/s
Network size	Modest	~50 devices per BAN
Frequency band	Global unlicensed and medical bands	MedRadio, ISM, WMTS, UWB
MAC	Scalable, reliable, versatile, self-forming	Low power, synchronization, listening, wake up, turn-around
QoS	Real-time data, periodic parametric data, episodic data and emergency alarms	P2P latency: from 10 ms to 250 ms, BER: from 10–10 to 10–3, reservation and prioritization
Coexistence	Coexistence with legacy devices and self-coexistence	Simultaneous co-located operation of up to 10 independent WBANs
Topology	Star, Mesh or Tree	Self-forming, distributed with multi-hop support
Environment	Body shadowing, attenuation	Seamless operation of multiple nodes in and out of scope with each other
Setup time	Not to be perceived	Up to 3 s
Security	Various levels	Authentication, Encryption, Authorization, Privacy, Confidentiality, Message integrity
Safety/Biocompatibility	Long-term continuous use without harmful effects	regulatory requirements
Ergonomic consideration	Size, weight, shape and form factors limited by location and organ	Non-invasive, appropriate size, weight and form factors
Reprogramming, Calibration, Customization	Personalized, configurable, integrated and context-aware services	reprogram, recalibrate, tune and configure devices wirelessly

**Table 5 sensors-22-03539-t005:** Non-invasive and invasive sensors.

Non-Invasive Sensors	Invasive/Implantable Sensors
EEG/ECG/EMG	Pacemaker
Position/Motion sensor	Deep brain stimulator
BP/SpO_2_	Implantable defibrillators
Glucose sensor	Cochlear implants
Temperature/Pressure sensor	Electronic pill for drug delivery
Pulse oximeter	Wireless capsule endoscope (electronic pill)
Oxygen, pH value	Retina implants

**Table 6 sensors-22-03539-t006:** Working mechanism of biosensors and their data rates in WBANs.

Sensor	Working Mechanism	Power Consumption	Data Rate
Blood sugar	Uses non-invasive methods such as optical measurement at the eye and breath analysis	Very low	Low
Blood pressure	Measures systolic and diastolic pressure	High	Low
ECG/EEG/EMG	Differential measurement via electrodes placed on the body	Low	High
Temperature	Uses an integrated circuit to detect the temperature changes by measuring resistance	Low	Very low
Respiration	Measures the dissolved oxygen in a liquid with two electrodes, a cathode and an anode covered by a thin membrane	Low	Low
Accelerometer	Measures the acceleration relative to freefall in three axes	High	High
Carbon dioxide	measures the gas absorption using infrared light	Low	Low
Gyroscope	Measures the orientation based on the principles of angular momentum	High	High
Pulse oximetry	Measures the changes of absorbance ratio by the red or infrared light passing through the fingertip or earlobe	Low	Low
Humidity	Measures the conductivity changes	Low	Very low

**Table 7 sensors-22-03539-t007:** Main specifications for WBAN systems.

Specifications	Requirements
Topology	Star or star mesh hybrid, bidirectional link
Devices	Number Typically 6, Up to 16
Data Rate	10 Kb/s–10 Mb/s
Range	>3 m with low data rate under IEEE Channel Model
PER	<10% with a link success probability of 95% overall channel conditions
Latency	<125 ms (medical), <250 ms (non-medical)
Reliability	<1 s for alarm, <10 ms for applications with feedback
Power Consumption	>1 year (1% LDC and 500 mAh battery), >9 h (always “on” and 50 mAh battery)
Coexistence	Less than 10 BANs in a volume of 6 m × 6 m × 6 m

**Table 8 sensors-22-03539-t008:** Medical body area networks’ operation bands.

Operation Bands	Frequency Range		Disadvantages	Application
>Medical device radio communications [12]	401–406, 413–419, 426–432, 438–444, 451–457 MHz		Limited bandwidth [70]	In-body and on-body
Human body communications (HBC)	5–50 MHz		Affected by the human posture and surroundings [71]	In-body [72] and on-body [13]
Medical implant communication service spectrum [70]	402–405 MHz		Limited bandwidth	In-body [13]
Wireless medical telemetry service	608–614, 1395–1400, 1427–1432 MHz		Limited bandwidth [70] Not harmonized globally or regionally [73]	On-body
Industrial, scientific and medical (ISM)	2360–2500 MHz	2360–2390 MHz	Not suitable for critical life situations due to coexistence with aeronautical mobile telemetry [74]	On-body
		2390–2400 MHz	Limited bandwidth	On-body
		2400–2500 MHz	Unlicensed WBAN, occupied by IEEE 802.15.6, Wi-Fi, Blue-tooth, ZigBee.	On-body
Ultra wideband (UWB)	3.1–10.6 GHz		Incomplete spectrum monitoring campaign [75]	On-body

**Table 9 sensors-22-03539-t009:** Descriptions of IEEE 802.15.6 channel models. Data from Ref. [82].

Scenario	Description	Frequency Band	Channel Model
S1	Implant to Implant	420-405 MHz	CM1
S2	Implant to Body Surface	420-405 MHz	CM2
S3	Body surface to Body Surface	13.5, 50, 400, 600, 900 MHz 2.4, 3.1–10.6 GHz	CM3
S4	Body Surface to External	900MHz, 2.4, 3.1–10.6 GHz	CM4

**Table 10 sensors-22-03539-t010:** The existing WBAN channel models.

Model Descriptions	Scenarios	Method	Propagation Effects	Mobility	Link Type
Dynamic channel model [84]	on-body, off-body, and body-to-body	finite-difference time-domain	fade variation and their corresponding amplitude distributions	walking	hand and thigh
A filter based probabilistic model [85]	Intra-WBAN	orthogonal frequency-division multiplexing	fading and dynamic variation challenges	static sitting and dynamic walking	hand
Simulations-based space-time dependent channel model [86]	Intra-WBAN, Indoor or Anechoic Chamber	Combination of frequency, distances in free space and around the body	Spatial and temporal characteristics-based fading. Shadowing due to body parts length and size.	Standing. walking and running	Hip to Wrist/Foot/Thigh. Arm to Foot and Head to Head
Measurement-based time-varying model [87]	Intra-WBAN, Indoor or Anechoic Chamber	Time-frequency and scenario-based	Slow and fast fading. Shadowing correlation between links.	Standing still, walking and running on the spot	Hip to Chest/right thigh/right wrist/right foot. etc.
Measurement and periodic characteristics-based model [88]	Intra-WBAN, Indoor or Anechoic	Distance and periodic function	Slow and fast fading along with Periodic Correlation	Standing, walking and running	Hip to Ankle/Wrist, Wrist to Wrist/Chest, Chest to Wrist/Hip
Simulation-based On and Off Body Multi antenna-channel model [89]	Intra-WBAN, Indoor	Geometrical-based statistical model	Multipath cluster of scatters	Walking	Head to Front/Back
IEEE proposed models [90]	Intra-WBAN. Indoor or Anechoic Chamber	Distance-based	Without spatial or temporal features	Static	Around torso and on-front part on the body

**Table 11 sensors-22-03539-t011:** Three security levels in the IEEE 802.15.6 standard for WBANs.

Security Levels	Protection Levels	Transmitted Frames
Level 0: lowest security level	Unsecured Communication	Data are transmitted in unsecured frames without encryption and authentication.
Level 1: medium security level	Authentication but no Encryption	Data are transmitted in plaintext form but secured authentication are involved.
Level 2: highest security level	Authentication and Encryption	Data are transmitted in secured authentication and encryption.

**Table 12 sensors-22-03539-t012:** Comparison of security techniques in WBANs.

Authors	Research Issues	Methodology	Outcome
Bengag et al. [112]	Jamming Attacks	Two MAC Protocols involved (ZIGBEE and TMAC)	Successful packet delivery rate
Arya et al. [113]	Data security	Constant monitoring for critical patients	Data authentication and authorization
Hayajneh et al. [114]	Lesser users	Increased storage level	More users and network lifetime
Thamilarasu et al. [115]	Network-level intrusion attacks	Machine learning and regression algorithms	Accurate results and lesser resource overhead
Umar et al. [116]	Active and passive network attacks	Enables mutual trust and used seed update algorithm	Minimal routing overhead and less computational cost
Dharshini et al. [117]	Vulnerable attacks	Secret key extraction with movement aided from DoS attacks	Minimum power consumption with high QoS
Suchithra et al. [118].	High-rate attacks	Maintain the bandwidth conditions in cooperative routing	Low-rate attacks
Kumar et al. [119].	Several security issues	Cloud technology and wireless communication	High storage and low computation cost
Rao et al. [120].	High residual power	Fuzzy logic technique	Secure and stable performance
Ali et al. [121].	User impersonation attacks	Bilinear pairing and elliptic curve cryptography	High security

**Table 13 sensors-22-03539-t013:** Performance comparison of the existing ASIC design methods.

Authors	Wang et al. [147]	Chen et al. [143]	Liu et al. [142]
Process technology (nm)	65	130	180
Modulation	DBPSK, DQPSK, D8PSK	DBPSK, DQPSK, D8PSK	BFSK ^a^
Power supply (V)	1.2	1.0	1.1
Core power of transmitter (μW)	1.69	9.89	34
Core power of receiver (μW)	20.46	—	39.6
Maximum throughput (Mbps)	10	0.97	0.625
Core size (mm^2^)	0.017	0.016 ^b^	0.31
Size of transmitter (mm^2^)	0.002	0.97	—
Size of receiver (mm^2^)	0.015	—	—

^a^ BPSK is not supported in the IEEE 802.15.6 standard. ^b^ It only contains the transmitter without the receiver.

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
