# Peer review of "Technological Requirements and Challenges in Wireless Body Area Networks for Health Monitoring: A Comprehensive Survey"

_sensors, 2022, doi:10.3390/s22093539_

Round 1

Reviewer 1 Report

This paper mainly introduces the applications, prospects and challenges of WBAN in the medical wearable field. A series of related contents from sensor classification to network protocol are expounded in detail. In addition, the paper also introduces an innovative research by the team aimed at improving security and data privacy, which is expected to promote the further development of WBAN.

  1. Line 82 of the article introduces that one of the tasks of the sensor is to transmit information to Tier-2. It is recommended to use the same technology as line 86 to transmit the collected information, and make a brief introduction.

  1. Now that 5G technology has gradually become popular, it is suggested that 5G technology can be added to the content of line 86 of the article.

  1. What is the classification basis of the typical application of WBAN mentioned in line 123 of the article? The first and third categories are distinguished according to the detection method of WBAN. The second category is generally summarized as remote control of medical devices. Is there a better one? , and the fourth row appears in the table, which conflicts with "three parts" in the text, and it is recommended to adjust the content of the table or text.

  1. Whether these typical applications in Table 1 can provide some examples or corroborate relevant literature to increase the persuasiveness of the article.

  1. The concluding introduction in line 125-129 lacks citations, and it is recommended to add relevant literature for support.

  1. Missing references on lines 144-145.

  1. The row spacing of Table4 can be slightly increased, and the content of the third column is difficult to distinguish.

  1. The spacing of the first column of Table5 is a bit strange, it is recommended to adjust it.

  1. The contents of Figure3 and Table5 overlap. It is recommended to add the contents of Table5 to Figure3, or Table5 to classify Figure3.

  1. Line 234, the table9 mentioned in the article was not found.

  1. 264 lines, neither does table10.

  1. Lines 304-308, some references can be added appropriately.

  1. Lines 314-377 basically have no citations, which is not reasonable.

  1. The line spacing at the beginning of line 368 is wrong.

  1. Line 378 , table11is missing.

  1. Line 386, table12 is missing.

  1. Starting from line 431, since the research of your team is mentioned, are there any related papers to cite.

  1. Line 482, table13 is missing.

  1. There are a lot of mentioned charts missing in the article. It is recommended to revise carefully and add the missing content.

Reviewer 2 Report

This paper is very well written, interesting and useful. The authors present the recent WBAN medical applications, existing requirements, challenges, solutions and technologies. The idea and the concept of the manuscript are very attractive for the “Sensors” journal community. This paper’s usefulness for researchers in the field is certain, as it contains engaging, helpful and practical results. Moreover, the paper is correctly formatted, with its sections appearing in the correct order, its figures and tables being clear and informative with the appropriate captions, and its references being numerous and properly cited in the text, while there are not any ethical issues involved.

Consequently, this manuscript is suitable for publication and applies to the high quality standards of the papers in the “Sensors” journal, since the addressed problem is important and the submitted paper will be a good reference to the “Sensors” researchers and readers.

Author Response

Thanks to the reviewer for your recognition of our manuscript.

Reviewer 3 Report

In this paper,  the authors review the WBAN applications in healthcare, focusing on various diseases monitoring, and the technical requirements and challenges. The paper has provided a good survey of  WBAN with corresponding descriptions on the specific areas where it is applicable and the general communication requirements of each of the different technologies. The organization structure of the paper is also not in bad shape. However, the paper misses several tables mentioned in the text (or references them in wrong manner) and the writing of some sections is not well done or overlooked. Some sections also need to have more details on their corresponding topics. Therefore I recommend major revision.

Some  comments:

- On page 13 the fonts are different than the rest of the text.
- Where is Table 12 as mentioned on page 13 in the text? "Table 12 shows the comparison of the recent security techniques in WBAN"
- Application-Specific Integrated Circuit (ASIC) has been unabbreviated in many places, e.g. on page 15. The abbreviation use should be double checked.
- Table 13 mentioned on page 15 is not present "demonstrated in Table 13."
- It is not clear what does Fig.7 and Fig. 8 demonstrate as scientific knowledge? 
- It is not clear the design of a specific ASIC  mentioned in the text improves the security and reduce energy consumption. There are no results on that side inside the manuscript. I would advice the authors to remove those section which is claimed to be the contributions of the authors which do not related well to the rest of the paper.
- I advice the authors to make a summary of each section at the end and also discuss some of the outcomes of the survey in a separate section as lessons learnt. 
- Does Table 9 belong to [62] on page 9?
- What does Section 4.3 title mean? "4.3. player Networking and Protocols" Is this section related to issues in WBAN?
- I think references to various tables should be fixed by introducing those tables inside the text as the paper claims to be a survey paper and the content should cover wide range of areas.
- Section 4.2 Channel Models should be extended. Mathematical formulas should also be introduced for WBAN channel models.
- On page 10 line 263, the authors mention "Several dynamic WBAN channel models41 have been proposed as shown in Table 10." Which Table 10?

Round 2

Reviewer 3 Report

The authors have responded to my previous comment. 

On page 17, I would not call the table title "Performance comparison between the proposed ASIC design and others" since the proposed ASIC design is already excluded from the paper.